# EnOF-SNN: Training Accurate Spiking Neural Networks via Enhancing the Output Feature

**Yufei Guo**[*], **Weihang Peng**[*], **Xiaode Liu, Yuanpei Chen, Yuhan Zhang, Xin Tong,**
**Zhou Jie, Zhe Ma**[†]

Intelligent Science & Technology Academy of CASIC

`yfguo@pku.edu.cn, pengweihang812@163.com, mazhe_thu@163.com`

## Abstract

Spiking neural networks (SNNs) have gained more and more interest as one of the energy-efficient alternatives of conventional artificial neural networks (ANNs). They exchange 0/1 spikes for processing information, thus most of the multiplications in networks can be replaced by additions. However, binary spike feature maps will limit the expressiveness of the SNN and result in unsatisfactory performance compared with ANNs. It is shown that a rich output feature representation (*i.e.*, the feature vector before classifier) is beneficial to training an accurate model in ANNs for classification. We wonder if it also does for SNNs and how to improve the feature representation of the SNN. To this end, we materialize this idea in two special designed methods for SNNs. First, inspired by some ANN-SNN methods that directly copy-paste the weight parameters from trained ANN with light modification to homogeneous SNN can obtain a well-performed SNN, we use rich information of the weight parameters from the trained ANN counterpart to guide the feature representation learning of the SNN. In particular, we present the SNN's and ANN's feature representation from the same input to ANN's classifier to product SNN's and ANN's outputs respectively and then align the feature with the KL-divergence loss as in knowledge distillation methods, called $\mathcal{L}_{\text{AF}}$ loss. It can be seen as a novel and effective knowledge distillation method specially designed for the SNN that comes from both the knowledge distillation and ANN-SNN methods. Second, we replace the last Leaky Integrate-and-Fire (LIF) activation layer as the ReLU activation layer to generate the output feature, thus a more powerful SNN with full-precision feature representation can be achieved but with only a little extra computation. Experimental results show that our method consistently outperforms the current state-of-the-art algorithms on both popular non-spiking static and neuromorphic datasets.

## 1 Introduction

Recently, convolutional neural networks (CNNs) have become extremely popular due to their performing more and more well in diverse fields including pattern recognition He et al. (2016); Simonyan & Zisserman (2014), object detection Girshick (2015); Ren et al. (2016); Ming et al. (2023), language processing Chen et al. (2016), robotics Levine et al. (2015), and so on. However, these full-precision CNN models make them power-hungry and tedious for real-world deployment. The spiking neural network (SNN), which aims to mimic the behavior of the human brain, has become a promising energy-efficient architecture to substitute for the CNN in some specific scenarios Guo et al. (2022c); Rathi et al. (2020); Zhu et al. (2022); Wu et al. (2019a); Zhang et al. (2020); Jeffares et al. (2021);

---

[*]Equal Contributions.

[†]Corresponding author.

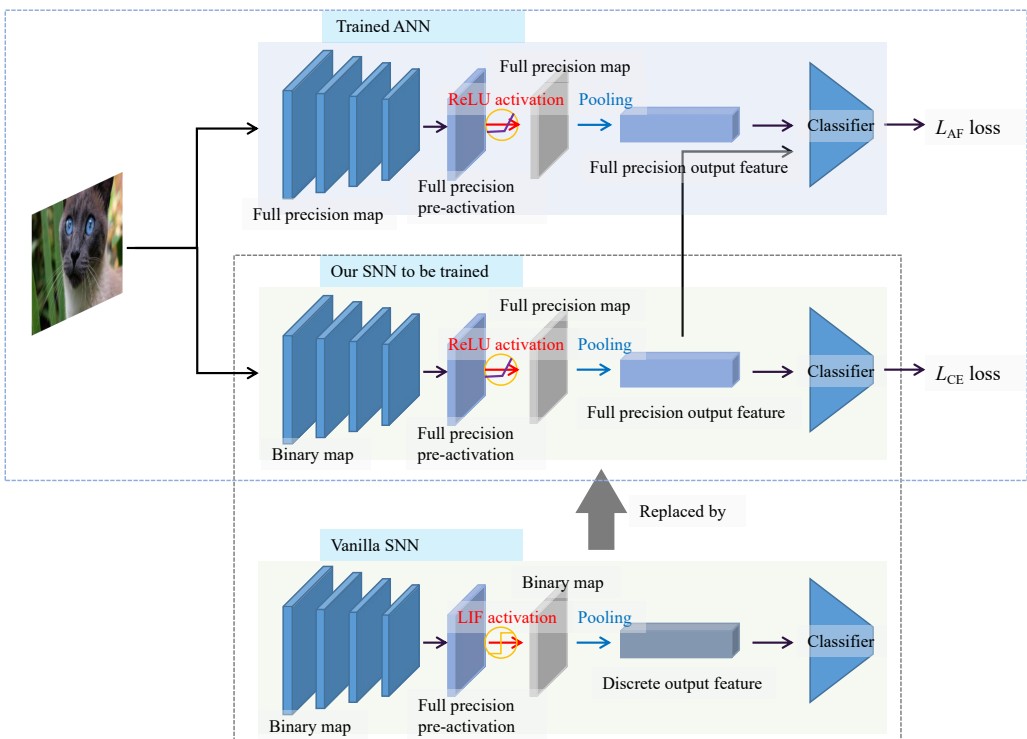

Figure 1: The overall workflow of the proposed method. To enhance the output feature representation, the $\mathcal{L}_{AF}$ loss is used to transmit the important information of output feature of the ANN to that of the SNN counterpart and the last LIF activation neuron layer is replaced by ReLU activation in our SNN structures.

Guo et al. (2022b, 2024b); Wang et al. (2023, 2022); Zhou et al. (2023). It adopts 0/1 spikes to transmit information. Benefitting from such an information processing paradigm, the multiplications of activations and weights of SNNs can be replaced by additions, thus enjoying low power consumption Guo et al. (2023a). Furthermore, on neuromorphic hardwares Akopyan et al. (2015), when there is no spike coming, the SNN neuron will remain inactive thus requiring significantly lower energy.

Despite the SNN being more energy-efficient compared with the ANN, it suffers unsatisfactory task performance. Binary spike feature maps of the SNN will result in limited expressiveness and large accuracy degradation compared with full-precision feature maps of the ANN Guo et al. (2023a, 2022a). Although the accuracy degradation problem can be mitigated by increasing the timesteps of the SNN, the inference burden will be increased too with the timestep increasing. It is shown crucial that output feature representation is rich and powerful for training a highly accurate model for the classification in recent studies in ANNs Chen et al. (2020); He et al. (2019); Kang et al. (2019). Here, we also expect that improving the output feature representation is beneficial for the accuracy of the SNN. To this end, we advocate using trained ANN's rich output feature to guide the learning of output feature representation of SNN. However, different from other SNN distillation methods that align the output feature of the SNN with that of the ANN counterpart directly, we adopt the trained ANN's classifier to guide the information transfer. Using the ANN's classifier directly is inspired by some ANN-SNN methods that directly copy-paste the weight parameters from trained ANN with light modification to homogeneous SNN can obtain a well-performed SNN Han et al. (2020); Li et al. (2021a). Therefore, we expect that utilizing the homogeneous ANN's some modules to guide the learning of SNN will be useful. In particular, for the same input, we wish the SNN's and ANN's feature representation to produce the same output when passed through the same ANN's classifier and the idea can be achieved by a simple Kullback-Leibler (KL)-divergence loss.

Furthermore, we also suggest that the last firing activation neuron layer in SNN could be replaced by ReLU, called RepAct, thus the output feature representation will be full precision and the expression ability will be improved but with a trivial additional cost compared with the vanilla SNN. It is worth

noting that spiking neural networks do have a certain number of non-binary feature maps. For example, many works use original images as the input of SNNs, thus the multiplications of the first layer can not be replaced by additions in SNNs Rathi & Roy (2020); Guo et al. (2022b); Baltes et al. (2023); Li & Zeng (2022), but the accuracy would boost to a high level. With a trivial additional cost, the RepAct can also improve the SNN greatly, hence it could be seen as a practical choice for many tasks too.

To sum up, we find that the expression ability of the output feature of the SNN is limited. Under this understanding, we focus on enhancing the output feature representation of the SNN in the paper. To this end, we propose two very simple but effective methods. First, we utilize the trained ANN to guide the learning of the output feature representation of the SNN. In specific, since the feature representation of the ANN is more powerful, we propose to align the output feature layer of SNN with that of ANN to learn its rich capabilities. Because direct feature representation matching does not take into account the classification task at hand and ignores the high-level representation information, thus will adversely affect SNN classification results if the matching strategy design is not good enough, we utilize the ANN's pre-trained classifier to align the output feature of the SNN with that of the ANN counterpart to enrich the SNN's feature representation, called $\mathcal{L}_{\mathrm{AF}}$ loss. This idea is based on the observation that directly copying-pasting the weight parameters from trained ANN to homogeneous SNN is always enough to obtain a well-performed SNN in ANN-SNN methods. In this sense, it can be seen as a simple and effective knowledge distillation method specially designed for SNN born in ANN-SNN. In this way, the feature representation and classification information can both be considered. Second, we replace the last LIF activation neuron layer with the ReLU activation layer to produce the full precision output feature with only a little extra computation, named RepAct. The overall workflow can be seen in Fig.1. Our contributions can be summarized as follows:

- We propose to use the well-trained ANN to guide and improve the learning of the output feature representation of the SNN. Along with taking into account the classification task, the $\mathcal{L}_{\mathrm{AF}}$ loss is proposed to weigh the discrepancy of the two outputs that come from the ANN's and SNN's feature representations for the same image by feeding in the same trained ANN's classifier. It can be seen as a simple and effective knowledge distillation method specially designed for SNN derived from a combination of advantages of knowledge distillation and ANN-SNN. With the $\mathcal{L}_{\mathrm{AF}}$ loss, the SNN's feature representation can be extremely enhanced and results in high accuracy.

- We also advise replacing the last LIF activation neuron layer in the SNN with ReLU activation layer to obtain the full precision output feature representation. This improvement will increase the expression ability of the output feature layer thus improving accuracy greatly, but with a trivial extra computation.

- We evaluate our methods on both famous static and spiking datasets. Extensive experimental results show that our method consistently outperforms state-of-the-art methods in various experimental settings.

## 2 Related Work

In view of the success of deep CNNs, the early development of deep SNNs benefits from the experience of CNNs. Using the trained ANN to guide the generation of high-precision SNN is one of the mainstream research at present. There are three main routes to realizing this idea.

First, a simple and effective way to use ANNs is converting a well-trained ANN model to an SNN model Han et al. (2020); Sengupta et al. (2018); Li et al. (2021a); Ding et al. (2021); Hao et al. (2023a,b). This method trains an accurate ANN with ReLU activation first and then uses the ANN network parameters to construct a special SNN network that enjoys the same structure as the ANN and adopts the average output through multiple timesteps as the final result. The converted SNN usually performs worse than the original ANN, since there are more or fewer conversion errors between the ANN and SNN. To bridge this gap as much as possible, many effective methods have been proposed, such as long inference time Girshick (2015), threshold rescaling Sengupta et al. (2018), soft reset Han et al. (2020), threshold shift Li et al. (2021a), and the quantization clip-floor-shift activation function Ding et al. (2021). However, the conversion method is restricted to rate-coding and the Integrate-and-Fire (IF) model, and needs long timesteps to obtain a satisfied accuracy.

Second, using the well-trained ANN model's parameters to initial the nascent SNN has been another fundamental line of research. For example, in Rathi et al. (2020) and Rathi & Roy (2020), a trained ANN is converted to an SNN first and then the weights of the converted SNN are fine-tuned further with the training method. Initialed with the trained ANN, the SNN shows a faster convergence than that from random initialization. Though this method usually needs fewer timesteps than the conversion method, it does not show obvious advantages for deep models.

Third, there are also some works that apply the knowledge distillation technique in the SNN field that to use ANN to distill the SNN Kushawaha et al. (2021); Takuya et al. (2021); Xu et al. (2023b,a); Guo et al. (2023c); Zhang et al. (2024). However, as has been comprehensively and systematically studied in Tian et al. (2020), most of the knowledge distillation methods perform poorly without well-designed hyper-parameters. These knowledge distillation methods for SNNs have also verified this consensus. They all adopt complex multi-stage distillation manner and only report the results on small-scale datasets. Providing a simple but effective knowledge distillation for the SNN is difficult.

Despite the above methods providing us with a lot of valuable insights, we still think there are some other improvements worth further consideration. First, these prior methods merely use the parameters of the trained ANN to guide the designing of the parameters of the SNN or adopt a complex framework to transfer knowledge from ANN to SNN individually, designing a simple and effective knowledge distillation method specifically for SNNs combining these useful experience of SNN knowledge distillation and ANN-SNN methods simultaneously are ignored. Second, these SNN methods all adopt similar architectures as ANNs that only modify the ANN activation layers as spike neuron layers simply but without considering the advantages and disadvantages of the SNNs comprehensively. In the paper, aiming at improving the output feature representation of the SNN, we will take care of these two problems well.

## 3 Preliminary and Methodology

This section first introduces the proposed method of aligning the SNN's output feature with the ANN's output feature. Then the Leaky Integrate-and-Fire (LIF) neuron model, why its output feature's expression ability is limited, and how to replace the LIF activation layer with the ReLU activation layer will be introduced in detail. Finally, some key details for training the SNN will be given.

### 3.1 $L_{\mathrm{AF}}$ Loss: Output Feature Alignment Loss

To improve the output feature representation ability, we advocate using the well-trained ANN to guide and improve the learning of the output feature representation of the SNN. A natural idea is to align the output feature of the SNN to that of its ANN counterpart directly like knowledge distillation methods, given by

$$\arg\min_{\mathbf{W}_{\mathrm{s}}}(||\boldsymbol{h}_{\mathrm{a}} - \boldsymbol{h}_{\mathrm{s}}||). \tag{1}$$

Where $\mathbf{W}_{\mathrm{s}}$, $\boldsymbol{h}_{\mathrm{a}}$, and $\boldsymbol{h}_{\mathrm{s}}$ represent the parameters of the SNN, the output feature of the ANN, and the output feature of the SNN.

However, we found that this natural idea does not always work, which has also been validated by Tian et al. (2020). The work has investigated most of the famous knowledge distillation methods and found that they perform poorly without well-designed hyper-parameters.

The reason why direct distillation performs worse in SNNs comes from that: 1) discrete output feature of the SNN makes its representation space much different from that of the full precision ANN, and directly aligning their output features simply will ignore the difference of their representation spaces and induce too much constraint in the SNN optimization, which may lead to a configuration far from the global optimal; 2) this design also treats each dimension in the feature space independently, and ignores the high-level interdependencies of the feature representation; and 3) it does not take into account the classification task at all, which conflicts with the final optimization goal.

To consider the above problems comprehensively, we present a simple but effective method, that uses the ANN's classifier to guide the learning of the SNN's output feature. This is inspired by some ANN-SNN methods that directly copy-paste the weight parameters from trained ANN to homogeneous SNN can obtain a well-performed SNN. Hence, we guess that utilizing a homogeneous ANN's some elements to guide the learning of SNN may be useful. In specific, we fed the SNN's output feature to

the ANN's pre-trained classifier to generate the output, $\boldsymbol{P}_\text{s}$. At the same time, we obtain the ANN's output, $\boldsymbol{P}_\text{a}$ from the same input image as the SNN. Then we utilize the KL-divergence loss to align the student output to the teacher output as follows:

$$\mathcal{L}_\text{AF} = \sum \texttt{SoftMax}(\boldsymbol{P}_\text{a})\log\left(\frac{\texttt{SoftMax}(\boldsymbol{P}_\text{a})}{\texttt{SoftMax}(\boldsymbol{P}_\text{s})}\right). \tag{2}$$

In this situation, since $\boldsymbol{P}_\text{s}$ and $\boldsymbol{P}_\text{a}$ come from the same ANN's pre-trained classifier, if $\boldsymbol{h}_\text{s}$ is same as $\boldsymbol{h}_\text{a}$, $\texttt{SoftMax}(\boldsymbol{P}_\text{s})$ will become same as the $\texttt{SoftMax}(\boldsymbol{P}_\text{a})$, then the $\mathcal{L}_{AF}$ will become zero and the smallest. At the same time, there are still other chances that can make $\mathcal{L}_{AF}$ small too. That is, $\mathcal{L}_{AF}$ will not lose the consideration that $\boldsymbol{h}_\text{s} = \boldsymbol{h}_\text{a}$, but will also take care of other alignment situations of $\boldsymbol{h}_\text{s}$ and $\boldsymbol{h}_\text{a}$. Furthermore, Eq. 2 also takes the classification task into the consideration. Hence, Eq. 2 can be seen as a more general alignment strategy than Eq. 1 designed specifically for SNNs. Finally, taking classification loss together, the total loss can be written as

$$\mathcal{L}_\text{Total} = \mathcal{L}_\text{CE} + \lambda\mathcal{L}_\text{AF}, \tag{3}$$

where $\mathcal{L}_\text{CE}$ is the cross-entropy loss, and $\lambda$ is a coefficient to balance the classification loss and the alignment loss.

## 3.2 RepAct: Replace the Last Activation Layer

As abovementioned, we argue that the last spike neuron layer will limit the performance of the SNN. To better depict this, we give the specific form of a spiking neuron first. Here, we adopt the well-known Leaky Integrate-and-Fire (LIF) neuron model for an SNN. The LIF neuron adjusts the membrane potential based on the input and its membrane potential at the previous moment as follows,

$$\boldsymbol{U}(t, \text{pre}) = \tau_\text{decay}\boldsymbol{U}(t-1) + \mathbf{W}\boldsymbol{X}(t), \tag{4}$$

where $\boldsymbol{U}(t, \text{pre})$ is the pre-membrane potential at $t$-th timestep, $\boldsymbol{U}(t)$ is the membrane potential at $t$-th timestep, $\tau_\text{decay}$ is the membrane time constant to describe the membrane potential decaying which is set as 0.25 in the paper, $\mathbf{W}$ is the weight, and $\boldsymbol{X}(t)$ is the binary map comes from the previous layer at $t$-th timestep. Since the $\boldsymbol{X}(t)$ is a binary tensor, $\mathbf{W}\boldsymbol{X}(t)$ can be realized by additions, which is more energy-efficient than multiplications.

Next, if $\boldsymbol{U}(t)$ exceeds a threshold, the LIF neuron will fire a spike and then hard-reset to 0, given by

$$\boldsymbol{O}(t) = \left\{ \begin{array}{ll} 1, & \text{if } \boldsymbol{U}(t, \text{pre}) \geq V_\text{th} \\ 0, & \text{otherwise} \end{array} \right. , \tag{5}$$

$$\boldsymbol{U}(t) = \boldsymbol{U}(t, \text{pre}) \cdot (1 - \boldsymbol{O}(t)), \tag{6}$$

where $V_\text{th}$ is a given firing threshold and $\boldsymbol{O}(t)$ is the output of the LIF neuron.

Let set $\boldsymbol{O}(t, \text{last})$ as the binary output of the last firing activation layer. It will be presented to the average-pooling layer and then full-connected layer, given by

$$\boldsymbol{y}(t) = \mathbf{W}_f\texttt{Avgpool}(\boldsymbol{O}(t, \text{last})), \tag{7}$$

where $\boldsymbol{y}(t)$ is the output of the network at $t$-th timestep and $\mathbf{W}_f$ is the weight matrix of the full-connected layer. obviously, the output feature, $\texttt{Avgpool}(\boldsymbol{O}(t, \text{last}))$ will be a discrete vector with a limited number of values. Hence, the representation ability of the output feature will be constrained compared to a full precision vector. We advocate for using the ReLU activation layer to replace the last LIF activation layer. Then the output of the last activation layer will be

$$\boldsymbol{O}(t, \text{last}) = \texttt{ReLU}(\mathbf{W}\boldsymbol{X}(t)). \tag{8}$$

Thus we will obtain a full precision output feature representation with a stronger expression ability. With the rich and powerful output feature, the SNN can converge to a highly accurate model easily but with a trivial cost.

## 3.3 Training Framework

We train the SNN with our method using the spatial-temporal backpropagation (STBP) algorithm Wu et al. (2019b), which treats the spiking neuron as a self-recurrent neural network thus enabling an

---

**Algorithm 1** Training SNN for one epoch.

---

**Input**: An SNN to be trained whose last LIF activation layer is replaced by the ReLU activation layer ; a pre-trained ANN; timestep: $T$; training dataset; total training iteration in one epoch $I_{\text{train}}$.
**Output**: The trained SNN.

1: **for** all $i = 1, 2, \ldots, I_{\text{train}}$ iteration **do**
2:     Get mini-batch training data and class label: $\boldsymbol{Y}^i$;
3:     Feed the mini-batch training data into the SNN and the ANN networks;
4:     Calculate the SNN output $\boldsymbol{O}_s^i(t)$ and the output feature $\boldsymbol{h}_s^i(t)$ of each time step;
5:     Compute classification loss $L_{\text{CE}} = \frac{1}{T} \sum_{t=1}^{T} \mathcal{L}_{CE}(\boldsymbol{O}_s^i(t), \boldsymbol{Y}^i)$;
6:     Calculate the ANN output $\boldsymbol{P}_a^i$;
7:     Present the $\boldsymbol{h}_s^i(t)$ to the ANN's classifier to product the output $\boldsymbol{P}_s^i(t)$ of each time step;
8:     Compute alignment loss $L_{AF} = \frac{1}{T} \sum_{t=1}^{T} \mathcal{L}_{\text{AF}}(\boldsymbol{P}_s^i(t), \boldsymbol{P}_a^i)$;
9:     Compute the total loss $\mathcal{L}_{\text{Total}}$ by Eq. 3;
10:     Backpropagation using STBP algorithm Wu et al. (2019b) and the SG method (see in Eq. 9), then update the SNN model parameters.
11: **end for**

---

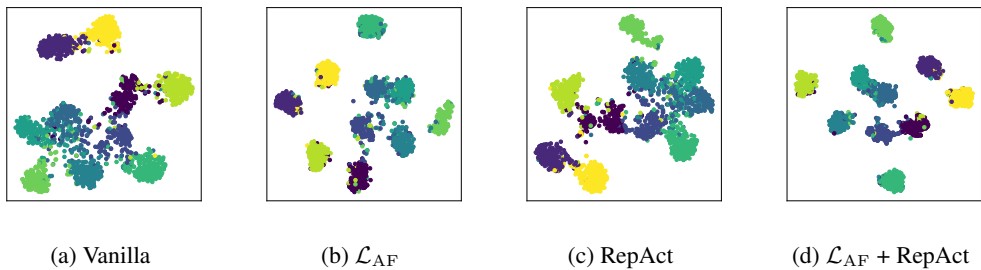

(a) Vanilla        (b) $\mathcal{L}_{\text{AF}}$        (c) RepAct        (d) $\mathcal{L}_{\text{AF}}$ + RepAct

Figure 2: t-SNE visualization of the output feature for random 2000 samples in CIFAR-10. Every color represents a different class. It can be clearly seen that our method can learn better representations.

error backpropagation mechanism for SNNs that follows the same principles as in conventional deep networks. As for the non-differentiable firing activity of the spiking neuron, we choose the STE surrogate gradients to solve it as doing in other surrogate gradient (SG) methods Rathi & Roy (2020); Guo et al. (2022b). Mathematically, it is defined as:

$$\frac{d\boldsymbol{O}}{d\boldsymbol{U}} = \begin{cases} 1, & \text{if } 0 \leq \boldsymbol{U} \leq 1 \\ 0, & \text{otherwise} \end{cases}. \tag{9}$$

Then, the SNN model can be trained end-to-end. As described in Zheng et al. (2020); Guo et al. (2022b, 2023b,d), with the SNN going deep, the distribution of membrane potential will shift accumulatively and fall into an inappropriate range, which will cause accuracy to decrease. To deal with this problem, we adopt the threshold-dependent Batch Normalization(tdBN) technique Zheng et al. (2020). The training algorithm of our method for one epoch is detailed in the Algo. 1.

## 4 Ablation Study

To understand how our method works in practice, a series of ablative studies for $\mathcal{L}_{\text{AF}}$ and RepAct with spiking ResNet20 architecture along with different timesteps were conducted on the CIFAR-10 dataset. In the experiment, we find that when $\lambda = 0.1$, $\mathcal{L}_{\text{AF}}$ can lead to a relatively better result and we adopt this setting. The top-1 accuracy of these models with $\mathcal{L}_{\text{AF}}$ and RepAct alone and their combinations are shown in Tab. 1. It's can be seen that both $\mathcal{L}_{\text{AF}}$ and RepAct can greatly improve the models' performances alone. furthermore, the performances can be still improved by combining these two methods. Take the first ablation experiment (*i.e.* timestep = 1) for example, the standard training of SNN is 90.40%. If we choose to use $\mathcal{L}_{\text{AF}}$ loss and RepAct alone, the performance would boost to 92.28% and 92.50%, which are huge improvements (more than 2.0%). Using the joint $\mathcal{L}_{\text{AF}}$

Table 1: Ablation experiments for RepAct and $\mathcal{L}_{AF}$.

| Timestep | Methods | Accuracy |
|---|---|---|
| 1 | baseline | 90.40% |
| | w/ $\mathcal{L}_{\mathrm{AF}}$ | 92.28% |
| | w/ RepAct | 92.50% |
| | w/ $\mathcal{L}_{\mathrm{AF}}$ & RepAct | **92.66%** |
| 2 | baseline | 92.80% |
| | w/ $\mathcal{L}_{\mathrm{AF}}$ | 93.53% |
| | w/ RepAct | 93.62% |
| | w/ $\mathcal{L}_{\mathrm{AF}}$ & RepAct | **93.86%** |
| 4 | baseline | 93.85% |
| | w/ $\mathcal{L}_{\mathrm{AF}}$ | 94.39% |
| | w/ RepAct | 94.66% |
| | w/ $\mathcal{L}_{\mathrm{AF}}$ & RepAct | **94.74%** |

and RepAct loss, we get a 2.26% accuracy improvement. Moreover, we also show the output feature representation ability of these SNN models with 1 timestep using t-SNE method van der Maaten & Hinton (2008) in Fig. 2. It can be observed that $\mathcal{L}_{\mathrm{AF}}$ and RepAct are both able to learn more discriminative features, which also correlates with quantitative accuracy gains.

# 5   Experiments

In this section, we conducted extensive experiments using widely-used spiking ResNet20 Rathi & Roy (2020); Sengupta et al. (2018), VGG16 Rathi & Roy (2020), ResNet18 Fang et al. (2021a), ResNet19 Zheng et al. (2020), and ResNet34 Fang et al. (2021a) to verify the effectiveness of the proposed methods on both static and neuromorphic datasets including CIFAR-10 Krizhevsky et al., CIFAR-100 Krizhevsky et al., ImageNet Deng et al. (2009), and CIFAR10-DVS Li (2017). The firing threshold $V_{\mathrm{th}}$ and the membrane potential decaying $\tau_{\mathrm{decay}}$ were set as $0.5$ and $0.25$ respectively. For static image datasets, we fed the images into the SNN model directly and used the first layer to encode the images to spikes, as in recent works Zheng et al. (2020); Rathi & Roy (2020). For the neuromorphic image dataset, we used the $0/1$ spike format directly. We report the top-1 accuracy results with the mean accuracy and standard deviation of 3 trials.

**CIFAR-10.** The CIFAR-10 dataset consists of 60K $32 \times 32$ images in 10 classes with 50K training images and 10K test images respectively. The data normalization, random horizontal flipping, cropping, AutoAugment Cubuk et al. (2019), and Cutout DeVries & Taylor (2017) were used for data augmentation as in Li et al. (2021b); Guo et al. (2022b). We utilized the SGD optimizer to train our models for 400 epochs with the 0.9 momentum and a learning rate of 0.1 cosine decayed to 0. We found that $\lambda = 0.1$ can lead to a relatively better result on the CIFAR dataset and we adopt this setting in the paper. From Tab. 3, it can be seen that our models achieve better performances than other SoTA methods for three commonly used networks in this field and can greatly reduce the timesteps, corresponding to inference time. Our VGG16 model with only 2 timesteps even outperforms the Diet-SNN Rathi & Roy (2020) with 10 timesteps by 1.31% accuracy. With only 2 timesteps, our method based on Resnet19 can also outperform the RecDis-SNN Zhang & Li (2020), the TET Deng et al. (2022), and the STBP-tdBN Zheng et al. (2020) with 6 timesteps by 0.64%, 1.69%, and 3.03% accuracy, respectively. The same superiority is also shown with the ResNet20 backbone. These comparison results clearly show the efficiency and effectiveness of our method.

**CIFAR-100.** We also verified our method on CIFAR-100. The CIFAR-100 dataset is similiar with the CIFAR-100 but in 100 classes. With the same training pipeline and the architectures on CIFAR-10 here, we found that our method can also achieves better accuracy than other prior works with fewer timesteps. Especially, the ResNet19 trained with our method can achieve 82.43% top-1 accuracy with timesteps of only 2, which outperforms the current methods, TET Deng et al. (2022) and TEBN Duan et al. (2022) by 7.71% and 6.02% but with 6 timesteps.

**ImageNet.** The ImageNet dataset consists of more than 1,250K training and 50K test images in 1000 classes with $224 \times 224$ respectively. We used standard data normalization, random horizontal flipping, and cropping for data augmentation. The SGD optimizer with the 0.9 momentum and a learning

Table 2: Comparison with SoTA methods on CIFAR-10.

| Dataset | Method | Type | Architecture | Timestep | Accuracy |
|---|---|---|---|---|---|
| CIFAR-10 | SpikeNorm Sengupta et al. (2018) | ANN2SNN | VGG16 | 2500 | 91.55% |
| | Hybrid-Train Rathi et al. (2020) | Hybrid training | VGG16 | 200 | 92.02% |
| | Spike-basedBP Lee et al. (2020) | SNN training | ResNet11 | 100 | 90.95% |
| | STBP Wu et al. (2019b) | SNN training | CIFARNet | 12 | 90.53% |
| | TSSL-BP Zhang & Li (2020) | SNN training | CIFARNet | 5 | 91.41% |
| | PLIF Fang et al. (2021b) | SNN training | PLIFNet | 8 | 93.50% |
| | GLIF Yao et al. (2022) | SNN training | ResNet19 | 2 | 94.44% |
| | DSR Meng et al. (2023) | SNN training | ResNet18 | 20 | 95.40% |
| | KDSNN Xu et al. (2023b) | SNN training | ResNet18 | 4 | 93.41% |
| | TAB Jiang et al. (2024) | SNN training | ResNet19 | 4 | 94.76% |
| | Diet-SNN Rathi & Roy (2020) | SNN training | VGG16 | 5 | 92.70% |
| | | | | 10 | 93.44% |
| | | | ResNet20 | 5 | 91.78% |
| | | | | 10 | 92.54% |
| | Dspike Li et al. (2021b) | SNN training | ResNet20 | 2 | 93.13% |
| | | | | 4 | 93.66% |
| | | | | 6 | 94.25% |
| | STBP-tdBN Zheng et al. (2020) | SNN training | ResNet19 | 2 | 92.34% |
| | | | | 4 | 92.92% |
| | | | | 6 | 93.16% |
| | TET Deng et al. (2022) | SNN training | ResNet19 | 2 | 94.16% |
| | | | | 4 | 94.44% |
| | | | | 6 | 94.50% |
| | RecDis-SNN Guo et al. (2022b) | SNN training | ResNet19 | 2 | 93.64% |
| | | | | 4 | 95.53% |
| | | | | 6 | 95.55% |
| | **Our method** | SNN training | ResNet19 | 1 | **95.37%**±0.08 |
| | | | | 2 | **96.19%**±0.10 |
| | | | ResNet20 | 1 | **92.66%**±0.08 |
| | | | | 2 | **93.86%**±0.07 |
| | | | | 4 | **94.74%**±0.08 |
| | | | VGG16 | 2 | **94.07%**±0.10 |
| | | | | 4 | **94.75%**±0.09 |

Table 3: Comparison with SoTA methods on CIFAR-100.

| Dataset | Method | Type | Architecture | Timestep | Accuracy |
|---|---|---|---|---|---|
| CIFAR-100 | RMP Han et al. (2020) | ANN2SNN | ResNet20 | 2048 | 67.82% |
| | BinarySNN Lu & Sengupta (2020) | ANN2SNN | VGG15 | 62 | 63.20% |
| | Hybrid-Train Rathi et al. (2020) | Hybrid training | VGG11 | 125 | 67.90% |
| | T2FSNN Park et al. (2020) | ANN2SNN | VGG16 | 680 | 68.80% |
| | TAB Jiang et al. (2024) | SNN training | ResNet19 | 4 | 76.81% |
| | Diet-SNN Rathi & Roy (2020) | SNN training | ResNet20 | 5 | 64.07% |
| | | | VGG16 | 5 | 69.67% |
| | Dspike Li et al. (2021b) | SNN training | ResNet20 | 2 | 71.68% |
| | | | | 4 | 73.35% |
| | | | | 6 | 74.24% |
| | TET Deng et al. (2022) | SNN training | ResNet19 | 2 | 72.87% |
| | | | | 4 | 74.47% |
| | | | | 6 | 74.72% |
| | RecDis-SNN Guo et al. (2022b) | SNN training | ResNet19 | 4 | 74.10% |
| | | | VGG16 | 5 | 69.88% |
| | Real Spike Guo et al. (2022c) | SNN training | ResNet20 | 5 | 66.60% |
| | | | VGG16 | 5 | 70.62% |
| | TEBN Duan et al. (2022) | SNN training | ResNet19 | 2 | 75.86% |
| | | | | 4 | 76.13% |
| | | | | 6 | 76.41% |
| | **Our method** | SNN training | VGG16 | 5 | **72.53%**±0.11 |
| | | | ResNet19 | 1 | **77.08%**±0.08 |
| | | | | 2 | **82.43%**±0.09 |
| | | | ResNet20 | 2 | **71.55%**±0.12 |
| | | | | 4 | **73.01%**±0.10 |

rate of 0.1 cosine decayed to 0 is adopted. Except that we set $\lambda = 1$ and epochs as 320, the other training settings are consistent with CIFAR dataset. Our results are presented in Tab. 4. Again, it can

Table 4: Comparison with SoTA methods on ImageNet.

| Dataset | Method | Type | Architecture | Timestep | Accuracy |
|---------|--------|------|--------------|----------|----------|
| ImageNet | STBP-tdBN Zheng et al. (2020) | SNN training | ResNet34 | 6 | 63.72% |
| | TET Deng et al. (2022) | SNN training | ResNet34 | 6 | 64.79% |
| | MS-ResNet Hu et al. (2021) | SNN training | ResNet18 | 6 | 63.10% |
| | OTTT Xiao et al. (2022) | SNN training | ResNet34 | 6 | 63.10% |
| | RecDis-SNN Guo et al. (2022b) | SNN training | ResNet34 | 6 | 67.33% |
| | SlipReLU Jiang et al. (2023) | ANN-SNN | ResNet34 | 32 | 66.61% |
| | Real Spike Guo et al. (2022c) | SNN training | ResNet18 | 4 | 63.68% |
| | | | ResNet34 | 4 | 67.69% |
| | SEW ResNet Fang et al. (2021a) | SNN training | ResNet18 | 4 | 63.18% |
| | | | ResNet34 | 4 | 67.04% |
| | OTTT Xiao et al. (2022) | SNN training | ResNet34 | 6 | 65.15% |
| | GLIF Yao et al. (2022) | SNN training | ResNet34 | 4 | 67.52% |
| | DSR Meng et al. (2023) | SNN training | ResNet18 | 50 | 67.74% |
| | **Our method** | SNN training | ResNet18 | 4 | **65.31%**$\pm$0.09 |
| | | | ResNet34 | 4 | **67.40%**$\pm$0.10 |

Table 5: Comparison with SoTA methods on CIFAR10-DVS.

| Dataset | Method | Type | Architecture | Timestep | Accuracy |
|---------|--------|------|--------------|----------|----------|
| CIFAR10-DVS | Rollout Kugele et al. (2020) | Rollout | DenseNet | 10 | 66.80% |
| | LIAF-Net Wu et al. (2022) | Conv3D | LIAF-Net | 10 | 71.70% |
| | LIAF-Net Wu et al. (2022) | LIAF | LIAF-Net | 10 | 70.40% |
| | STBP-tdBN Zheng et al. (2020) | SNN training | ResNet19 | 10 | 67.80% |
| | RecDis-SNN Guo et al. (2022b) | SNN training | ResNet19 | 10 | 72.42% |
| | Real Spike Guo et al. (2022c) | SNN training | ResNet19 | 10 | 72.85% |
| | | | ResNet20 | 10 | 78.00% |
| | Ternary Spike Guo et al. (2024a) | SNN training | ResNet19 | 10 | 78.40% |
| | | | ResNet20 | 10 | 78.70% |
| | **Our method** | SNN training | ResNet19 | 10 | **80.10%**$\pm$0.20 |
| | | | ResNet20 | 10 | **80.50%**$\pm$0.20 |

be observed that our method also reaches the SoTA. Our ResNet18 and ResNet34 achieve 65.31% and 67.40% top-1 accuracy with only 4 timesteps, which is just a little worse than GLIF Yao et al. (2022). However, GLIF adopts the complex spiking neuron, which will increase the inference cost. It's worth noting that our models even outperform better than these SEW ResNet-based models Fang et al. (2021a); Guo et al. (2022c), which transmits information with arbitrary integers. This shows the ability of our method to handle these large-scale datasets.

**CIFAR10-DVS.** The neuromorphic dataset, CIFAR10-DVS, was also used in the paper. It is the neuromorphic version of a part of the CIFAR-10 dataset which consists of 10K images in 10 classes. We also adopted the principle to split the dataset into 9K training images and 1K test images, and resize them to $48 \times 48$ to evaluate the models' performance similar to Wu et al. (2019b); Guo et al. (2022c,b). the random horizontal flip and random roll within 5 pixels are also adopted in the paper for augmentation Guo et al. (2022b). To train the ANN model on the neuromorphic dataset, we change the channel number of the first layer of the ANN to 20 to contain all the temporal input at once. Other training settings are the same as that for CIFAR-10 dataset. Our method reaches to 80.10% and 80.50% for ResNet19 and ResNet20.

# 6  Conclusion

This work aims at enhancing the output feature representation of SNNs. Then, the $\mathcal{L}_{\mathrm{AF}}$ Loss that improves the learning of the output feature representation of the SNN using the well-trained ANN classifier and the RepAct method which replaces the last LIF activation layer with the ReLU activation layer to generate a more powerful output feature are proposed. A series of ablation studies show that the two proposed methods can greatly increase the SNN's accuracy. The proposed methods can also be combined and will consistently outperform the other state-of-the-art methods.

## Acknowledgment

This work is supported by grants from the National Natural Science Foundation of China under contracts No.12202412 and No.12202413.

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
