# OpenReview forum: "EnOF-SNN: Training Accurate Spiking Neural Networks via Enhancing the Output Feature"
_NeurIPS.cc/2024/Conference — NeurIPS 2024 poster_

### Official Review · Reviewer_BFUv · 2024-06-26

**Soundness:** 3
**Presentation:** 3
**Contribution:** 3
**Rating:** 6
**Confidence:** 5

**Summary:**

The paper presents the method to improve the representation power of SNNs. To this end, the paper proposed two new techniques to the training of spiking neural networks: 1) To guide learning of the last feature layer by an alignment loss with a pre-trained non-spiking network and 2) to replace the last spiking LIF layer with a continuously valued ReLU layer.

**Strengths:**

1)Those two techniques are new and interesting and lead to new SOTA on all the datasets they try.
2)The paper is well-written.
3)The paper provided ablation analysis to evaluate the contributions of both proposed techniques.

**Weaknesses:**

1)	How can SNN representation be converted to float activation in eq 1? Is it averaged?
2)	Improvement with technique 2 is expected, and it makes the comparison with pure SNNs unfair.
3)	The method needs to train an ANN in addition to the SNN. So the total training time is longer.

**Questions:**

Please see weaknesses.

**Limitations:**

Please see weaknesses.

---

> ### Author Rebuttal · Authors · 2024-08-07
>
> Thank you for your efforts in reviewing our paper and your recognition of our good writing, technical soundness experimental results, and novel methods. The response to your questions is given piece by piece as follows.
>
> **W1**: How can SNN representation be converted to float activation in eq 1? Is it averaged?
>
> **A1**: Sorry for the confusion. Yes, it is averaged through timesteps. We will further clarify this as per your suggestion in the revised version. Thanks again.
>
> ---
>
> **W2**: Improvement with technique 2 is expected, and it makes the comparison with pure SNNs unfair.
>
> **A2**: Sorry for the confusion. Though the technique 2 will increase the energy consumption， it is very simple and effective. In the appendix, we show that the method just brings about a 1.0% energy consumption uplift. Thus the additional energy consumption could be ignored. In this sense, the comparison could be seen as fair.
>
> ---
>
> **W3**: The method needs to train an ANN in addition to the SNN. So the total training time is longer.
>
> **A3**: Sorry for the confusion. The additional training cost is trivial. We take the ResNet20 with 4 timesteps on CIFAR10 to compare. **The training time of ANN is about 6.25% of that of SNN on the same GPU**, as follows: since the memory consumption of SNN is 4 times that of ANN, ANN can use a batch size that is 4 times that of SNN, which means that the number of iterations required for ANN training is 25% of that of SNN. In addition, since timestep affects SNN, the training time of ANN in the same iteration is 25% of that of SNN. Overall, the training time of ANN is 6.25% of that of SNN. We report the training cost for 400 epochs running using spiking ResNet20 (batch=128) on CIFAR10 for our methods and the training cost for 100 epochs running using ResNet20 (batch=512) with the same single V100 here.  **The costs are 8348.87s and 561.77s. It can be seen that the additional training cost is trivial.**

---

> > ### Comment · Reviewer_BFUv · 2024-08-08
> >
> > I have reviewed the rebuttal. The revised version should clearly detail the average operation and clarify the comparisons to enhance readability.
> >
> > Overall, most concerns have been addressed. I would like to adjust the score to a weak accept.

---

> > > ### Author Response · Authors · 2024-08-11
> > > **thanks**
> > >
> > > Thanks very much for your reply and recognition. We are happy to see that your concerns have been addressed.

---

### Official Review · Reviewer_4HWX · 2024-07-07

**Soundness:** 3
**Presentation:** 3
**Contribution:** 3
**Rating:** 7
**Confidence:** 5

**Summary:**

The authors propose a novel distillation approach (EnOF) to address the mismatch in output precision between SNN and ANN, which leads to poor distillation performance. EnOF feeds the output feature of SNN into the ANN classification head to obtain its effective mapping output P_s, and then uses distillation learning to make the P_s close to the ANN output P_a. Additionally, the paper analyzes the issue of inadequate SNN representation due to the final LIF activation layer. The authors suggest replacing it with the ReLU activation function. Through the methods presented in this paper, SNN demonstrates significant performance improvements across different datasets.

**Strengths:**

The method proposed in this paper is simple, effective, and easy to implement. This paper presents a novel knowledge distillation learning concept for two different types of networks whose output representation abilities are different. I believe this article has a certain promoting effect on the SNN training.

**Weaknesses:**

To illustrate the advances made by EnOF and RepAct, the authors can provide the network Hessian matrix eigenvalues or convergence speed comparisons.

An intuitive drawback of RepAct is its requirement for high-precision floating-point matrix multiplication operations in the classification head, which undoubtedly increases the computation overhead. I suggest that the authors move the overhead comparison section from the appendix to the main text.

The main text lacks the definition of the method name "EnOF.”.

**Questions:**

1. Does distillation learning have a temperature coefficient?
2. Is the optimal $\lambda$ obtained through search, and does different $\lambda$ have a significant impact on the final results?

---

> ### Author Rebuttal · Authors · 2024-08-07
>
> Thank you for your efforts in reviewing our paper and your recognition of our simple but effective method. The response to your questions is given piece by piece as follows.
>
> ---
>
> **W1**: To illustrate the advances made by EnOF and RepAct, the authors can provide the network Hessian matrix eigenvalues or convergence speed comparisons.
>
> **A1**: Thanks for the advice. We will add the convergence speed comparisons as per your suggestion in the revised version. Thanks again.
>
> ---
>
> **W2**: I suggest that the authors move the overhead comparison section from the appendix to the main text.
>
> **A2**: Thanks for the advice. We will add the overhead comparison section to the main text as your suggestion in the revised version. Thanks again.
>
> ---
>
> **W3**: The main text lacks the definition of the method name "EnOF.”.
>
> **A3**: Sorry for the confusion. “EnOF” comes from the first letter of “**Enhancing the Output Feature**” from the title. We will further clarify it in the revised version. Thanks again.
>
> ---
>
> **Q1**: Does distillation learning have a temperature coefficient?
>
> **A1**: Sorry for the confusion. The distillation learning has a temperature coefficient. We set it as 0.1 for CIFAR dataset and 1 for ImageNet dataset.
>
> ---
>
> **Q2**: Is the optimal $\lambda$ obtained through search, and does different $\lambda$ have a significant impact on the final results?
>
> **A2**: Thanks for the question. The hyperparameter we used in the paper is 0.1 for CIFAR dataset and 1 for ImageNet dataset.  We get it just from a simple search strategy that chooses several hyperparameters and compares their results. Here we also report the results of 0.5 and 1.0 for CIFAR10. It can be seen that the hyperparameter will affect the results. Nevertheless, Using the KD is better than the Baseline.
>
> | Dataset | Method | Timestep | hyperparameter=0.1 | hyperparameter=0.5 | hyperparameter=1.0 |
> | --- | --- | --- | --- | --- | --- |
> | CIFAR10 | Baseline | 1 |  90.40% |  90.40% |  90.40% |
> |  | Ours | 1 | 92.28% | 91.96% | 91.78% |
> |  | Baseline | 2 |  92.80% | 92.80% | 92.80% |
> |  | Ours | 2 |  93.53% | 93.13% | 92.99% |
>
> ---

---

> > ### Comment · Reviewer_4HWX · 2024-08-09
> >
> > Thank you for your response. The authors have made efforts to enhance the readability of the manuscript, and I believe that this paper will contribute positively to the development of the SNN field. I'd like to raise my score to 7.

---

> > > ### Author Response · Authors · 2024-08-11
> > > **thanks**
> > >
> > > Thanks very much for your reply and recognition. We are happy to see that your concerns have been addressed.

---

### Official Review · Reviewer_8VEP · 2024-07-10

**Soundness:** 3
**Presentation:** 3
**Contribution:** 2
**Rating:** 5
**Confidence:** 4

**Summary:**

This paper introduces methods to improve the output feature representation of Spiking Neural Networks (SNNs) by utilizing knowledge distillation (KD) techniques and modifying activation functions. The approach involves aligning the SNN's output features with those of a pre-trained Artificial Neural Network (ANN) using KL-divergence loss and replacing the last Leaky Integrate-and-Fire (LIF) activation with a ReLU activation to enhance feature precision. These modifications are shown to increase the expressiveness and performance of SNNs on both static and neuromorphic datasets.

**Strengths:**

1. __Technical Soundness__: The experimental results support the effectiveness of the proposed methods, albeit not conclusively when considering potentially more efficient alternatives.

2. __Clarity__: The paper is well-written with a clear structure. The methods are described in detail, making it easy for readers to understand the proposed enhancements.

3. __Energy Estimation__: The discussion on energy estimation provides valuable insights into the practical implications of the proposed methods, highlighting their potential for energy-efficient computing.

**Weaknesses:**

1. __Novelty__: The methods, while integrating known techniques in a new context, do not present strong novelty. While the combination of techniques is original, the individual components (knowledge distillation and activation function manipulation) are well-trodden areas in SNNs. The paper could benefit from a deeper exploration of how these specific adaptations contribute uniquely to the SNNs' context.

2. __Citation Format__: The paper does not adhere to NeurIPS citation format requirements.

3. __Comparative Analysis__: The paper lacks a comprehensive comparison with other SNN training methods that use similar KD approaches, which is crucial for substantiating the claimed improvements. A broader comparison with similar approaches in the field of SNNs+KD (see [1, 2]) could better highlight this.

4. __Significance__: The improvements, while statistically valuable, may not represent a paradigm shift in SNN training that the field might expect for a high-impact publication.

**Questions:**

+ Could the authors extend their comparisons to include seminal works in SNN+KD such as those by Guo et al. [1] and Deng et al. [2]? This would help in positioning their method more clearly within the current research landscape.

+ The ablation study in Table 6 is critical, and the details of the hyperparameters used in the vanilla KD need to be listed. Given that the paper mentions that "most knowledge distillation methods perform poorly without well-designed hyper-parameters," the design of KD should be further discussed. For example, the inclusion of a temperature coefficient commonly used in logits-based KD [3] should be considered (in Eq. 2, etc) to potentially improve the alignment strategy.


Reference:

[1] Guo, Yufei, et al. "Joint a-snn: Joint training of artificial and spiking neural networks via self-distillation and weight factorization." Pattern Recognition 142 (2023): 109639.

[2] Deng, Shikuang, et al. "Surrogate module learning: Reduce the gradient error accumulation in training spiking neural networks." International Conference on Machine Learning. PMLR, 2023.

[3] Hinton, Geoffrey, Oriol Vinyals, and Jeff Dean. "Distilling the knowledge in a neural network." arXiv preprint arXiv:1503.02531 (2015).

**Limitations:**

See Weaknesses and Questions.

---

> ### Author Rebuttal · Authors · 2024-08-07
>
> Thank you for your efforts in reviewing our paper and your recognition of our good writing and technical soundness experimental results. The response to your questions is given piece by piece as follows.
>
> **W1**: The paper could benefit from a deeper exploration of how these specific adaptations contribute uniquely to the SNNs' context.
>
> **A1**: Thanks for your advice.  The starting point of our research is that a rich output feature representation is important for improving the accuracy of SNN first. Based on this perspective, we propose the L_{AF} Loss and RepAct.
>
> For the L_{AF} Loss, it is not merely a distillation method. The primary motivation behind its proposal is that many ANN-SNN works do not alter the weights of neural networks yet achieve high accuracy. This insight has inspired us to incorporate certain ANN modules into SNN. However, direct migration and training are incompatible. Hence, we introduce the concept of utilizing the classifier of ANN to optimize the output feature of SNN. We have chosen to formalize our idea through distillation and, in the supplementary materials, we provide evidence of its significant advantages over direct distillation. Thus, from a systemic standpoint, the method we propose is a distillation approach specifically tailored for SNN, which fundamentally differs from simple distillation despite their superficial similarities.
>
> For the RepAct, this method is very simple but effective. Maybe it is not very novel, but will provide a novel insight for the SNN field that enhancing the output feature is very useful for accuracy.
>
> We will further clarify this in the revised version as your suggestion. Thanks.
>
> ---
>
> **W2**: The paper does not adhere to NeurIPS citation format requirements.
>
> **A2**: Thanks for the reminder. We will further recheck and polish our paper in the final version.
>
> ---
>
> **W3**: The paper lacks a comprehensive comparison with other SNN training methods that use similar KD approaches.
>
> **A3**: Thanks for your advice. Here we add some comparison results for the same models with [1] and [2] based on their reported results. It can be seen that our method works well on ImageNet with ResNet-18 and on CIFAR10 with ResNet-19, but a little worse on CIFAR10 with ResNet-18. We will report more comparisons in the revised version. Thank you.
>
> | Dataset | Model | Timestep | Accuracy |
> | --- | --- | --- | --- |
> | CIFAR10 | Our ResNet-19 | 2 | 96.19% |
> |  | Our ResNet-18 | 2 | 93.86% |
> |  | Surrogate Module ResNet-19 [2] | 4 | 95.58% |
> |  | Surrogate Module ResNet-18 [2] | 2 | 94.58% |
> |  | Joint ResNet-18 [1] | 2 | 94.01% |
> | ImageNet | Our ResNet-18 | 4 | 65.31% |
> |  | Surrogate Module ResNet-18 [2] | 4 | 64.53% |
>
> ---
>
> **W4**: The improvements, while statistically valuable, may not represent a paradigm shift in SNN training that the field might expect for a high-impact publication.
>
> **A4**: Sorry for the confusion.  We think our most valuable contribution is providing insight for the SNN field that enhancing the output feature is much useful for accuracy. Our $L_{AF}$ and RepAct both are means, while our intention is to improve the output feature and prove that it is beneficial for boosting accuracy.  With verified experiments and under the perspective, we believe this work is much beneficial for the following SNN work. And many other methods to improve the output feature would be proposed, like special temporal contrastive loss or simply increasing channels for SNNs. Yet, this is not to say our methods are not novel. For RepAct, it is very simple, but effective. With a trivial additional cost, it can improve the SNN greatly. From a practical perspective, this is a really good choice for the industry. For $L_{AF}$ it is a simple and effective KD method designed for SNN, different from these straightforward KDs. In specific, we introduce its novelty in two-fold. First, providing a simple but effective KD for SNNs is difficult. Most of KDs perform poorly without well-designed hyper-parameters, which have been comprehensively and systematically studied in [1] (see Tab.1 and Tab.2 in [1]. This paper gets more than 700 cites and 1.8k stars). Most KD methods only work well on small datasets, especially in the SNN field.  Second, our $L_{AF}$ comes from both the KDs and ANN-SNN methods.  Some ANN-SNN methods that directly copy-paste the weight parameters from trained ANN to homogeneous SNN with minor changes can obtain a well-performed SNN. This inspires us that utilizing homogeneous ANN's some elements to guide the learning of SNN may be useful. Along with considering the classification and representation, we utilize ANN’s pre-trained classifier to train the SNN’s feature and it works well. Thus, we think $L_{AF}$ is new, simple, designed specifically for SNNs.
>
> [1] Tian, Y., Krishnan, D., and Isola, P. Contrastive representation distillation. In *International Conference on Learning Representations*, 2020.
>
> ---
>
> **Q1**: Could the authors extend their comparisons to include seminal works in SNN+KD such as those by Guo et al. [1] and Deng et al. [2]?
>
> **A1**: Thanks for your advice. Please see our response for the **W3**.
>
> ---
>
> **Q2**: The ablation study in Table 6 is critical, and the details of the hyperparameters used in the vanilla KD need to be listed.
>
> **A2**: Thanks for your advice. The hyperparameter we used in ablation is 0.1.  Here we also report the results of 0.5 and 1.0. It can be seen that the hyperparameter will affect the results. However, the effect is smaller on ours.
>
> | Dataset | Method | Timestep | hyperparameter=0.1 | hyperparameter=0.5 | hyperparameter=1.0 |
> | --- | --- | --- | --- | --- | --- |
> | CIFAR10 | Baseline | 1 |  90.40% |  90.40% |  90.40% |
> |  | Direct distillation | 1 | 91.59% | 91.03% | 90.63% |
> |  | Ours | 1 | 92.28% | 91.96% | 91.78% |
> |  | Baseline | 2 |  92.80% | 92.80% | 92.80% |
> |  | Direct distillation | 2 |  92.59% | 92.01% | 91.77% |
> |  | Ours | 2 |  93.53% | 93.13% | 92.99% |
>
> ---

---

### Official Review · Reviewer_bdtv · 2024-07-14

**Soundness:** 3
**Presentation:** 3
**Contribution:** 3
**Rating:** 7
**Confidence:** 4

**Summary:**

This paper explores whether the benefits of rich output feature representations, known to enhance the accuracy of ANN models for classification, also apply to SNNs. If so, the authors seek to improve the feature representation of SNNs.

The authors address this problem in two steps: first, they align the SNN output features with the ANN output features using KL-divergence loss, similar to knowledge distillation. Second, they replace the last Leaky Integrate-and-Fire (LIF) activation layer in the SNN with a ReLU activation layer.

For feature alignment, the same input is fed to both the SNN and the ANN, and their outputs are compared and aligned using KL-divergence loss. The idea is to use the trained ANN's rich output features to guide the learning of the SNN's output feature representation.

The proposed method involves training the ANN counterpart (with ReLU activation), then replacing the LIF with the ReLU activation function in the last layer of the SNN. Finally, the modified SNN is trained with the help of the pre-trained ANN counterpart by incorporating the KL loss into the total loss of the SNN.

**Strengths:**

1. This paper uses the well-trained ANN to guide and improve the learning of the output feature representation of the SNN. The KL-divergence loss (L_{AF} loss) is used to measure the discrepancy of the ANN's and SNN's feature representations when feeding in the same image into the two networks.

2. By replacing the last LIF activation neuron layer in the SNN with ReLU activation layer, the SNN shows a large improvement.

**Weaknesses:**

1. The proposed method requires training the ANN counterpart first and setting it aside as a pre-trained model. Subsequently, the SNN needs to be trained. This approach essentially involves training the same framework twice, once with ANN activations and once with SNN activations.

2. The method includes replacing the last LIF activation neuron layer in the SNN with a ReLU activation layer, which results in significant improvement. However, it is unclear whether the final improvement is primarily due to this replacement of last-layer's activation function or the feature alignment process.

**Questions:**

1. In this paper, the authors try to use rich information of the weight parameters from a pretrained ANN counterpart to guide the feature representation learning of the SNN.
Does this method require training the ANN counterpart and using it to assist in the subsequent training of the SNN? If there is true, it seems the paper trains both the ANN counterpart and the SNN, the training resources will be doubled, how to solve this problem?


2. The authors stated that they "replace the last Leaky Integrate-and-Fire (LIF) activation layer as the ReLU activation layer in the SNN". Does the improvement mainly come from this replacement of LIF with ReLU? From results of the Ablation experiments in table 1, it shows that using "w/ RepAct" (92.50) demenstrates a larger improvement than using "w/ L_{AF}" (92.28) compared to the "baseline" (90.40) for timestep=1. For other timestep (2 and 4), the results show the same trends. Moreover, the final method using "w/ L_{AF} & RepAct" achieved accuracy of 92.66, which is only a very small increase compared to the "w/ RepAct" with 92.50.


3. In order to check the improvement for the applications, can the authors provide the performance of "w/ RepAct" and "w/ L_{AF}", "w/ L_{AF} & RepAct", the baseline, and the ANN-counterpart for CIFAR10, CIFAR100, ImageNet, CIFAR10-DVS datasets?



4. Grammer errors, for example Line 42-44, "Despite the SNN being more energy-efficient compared with the ANN, it suffers unsatisfactory task performance, due to binary spike feature maps of the SNN will result in limited expressiveness and large accuracy degradation compared with full-precision feature maps of the ANN",

**Limitations:**

As the above Weaknesses and Questions.

---

> ### Author Rebuttal · Authors · 2024-08-07
>
> Thank you for your efforts in reviewing our paper and your recognition of our effective method and good results. The response to your questions is given piece by piece as follows.
>
> ---
>
> **W1**: The proposed method requires training the ANN counterpart first and setting it aside as a pre-trained model. Subsequently, the SNN needs to be trained. This approach essentially involves training the same framework twice, once with ANN activations and once with SNN activations.
>
> **A1**: Sorry for this confusion. The additional training cost is trivial. We take the ResNet20 with 4 timesteps on CIFAR10 to compare. **The training time of ANN is about 6.25% of that of SNN on the same GPU**, as follows: since the memory consumption of SNN is 4 times that of ANN, ANN can use a batch size that is 4 times that of SNN, which means that the number of iterations required for ANN training is 25% of that of SNN. In addition, since timestep affects SNN, the training time of ANN in the same iteration is 25% of that of SNN. Overall, the training time of ANN is 6.25% of that of SNN. We report the training cost for 400 epochs running using spiking ResNet20 (batch=128) on CIFAR10 for our methods and the training cost for 100 epochs running using ResNet20 (batch=512) with the same single V100 here.  **The costs are 8348.87s and 561.77s. It can be seen that the additional training cost is trivial.**
>
> ---
>
> **W2**: The method includes replacing the last LIF activation neuron layer in the SNN with a ReLU activation layer, which results in significant improvement. However, it is unclear whether the final improvement is primarily due to this replacement of last-layer's activation function or the feature alignment process.
>
> **A2**: Sorry for the confusion. From the ablation study, it can be seen that both the RepAct method and the alignment method can gain huge improvements alone.  However, the accuracy upper of a network is limited, and the difficulty will increase quickly with the accuracy reaching the upper. Thus using them both can not get twice the profit. The same phenomenon can also be seen in many works, e.g., IM-Loss (NeurIPS2022), using IM-Loss and ESG simultaneously would not bring much gain compared with a single.
>
> Guo Y, Chen Y, Zhang L, et al. IM-loss: information maximization loss for spiking neural networks[J]. Advances in Neural Information Processing Systems, 2022, 35: 156-166.
>
> ---
>
> **Q1**:  it seems the paper trains both the ANN counterpart and the SNN, the training resources will be doubled, how to solve this problem?
>
> **A1**: Sorry for this confusion. Please see our response to **W1**.
>
> ---
>
> **Q2**:  Does the improvement mainly come from this replacement of LIF with ReLU?
>
> **A2**: Sorry for this confusion. Please see our response to **W2**.
>
> ---
>
> **Q3**:   In order to check the improvement for the applications, can the authors provide the performance of "w/ RepAct" and "w/ L_{AF}", "w/ L_{AF} & RepAct", the baseline, and the ANN-counterpart for CIFAR10, CIFAR100, ImageNet, CIFAR10-DVS datasets?
>
> **A3**: Thanks for the advice. Here we add more experiments as your suggestion. Note that we use the ResNet20 architecture on CIFAR10 (with 4 timestep), CIFAR100 (with 4 timestep), and CIFAR10-DVS (with 10 timestep) and ResNet18 architecture on ImageNet (with 4 timestep).
>
> | Model\Dataset | CIFAR10 | CIFAR100 | ImageNet | CIFAR10-DVS |
> | --- | --- | --- | --- | --- |
> | ANN | 95.91% | 75.14% | 71.04% | 81.00% |
> | Vanilla SNN  | 93.85% |  71.77% | 61.07% | 78.70% |
> | Ours w/ RepAct | 94.66% | 72.63% | 63.78% | 79.63% |
> | Ours w/ L_{AF} | 94.39% | 72.98% | 64.86% | 79.34% |
> | Ours with both  | 94.74% | 73.01% | 65.31% | 80.50% |
>
> ---
>
> **Q4**:   Grammer errors, for example Line 42-44, "Despite the SNN being more energy-efficient compared with the ANN, it suffers unsatisfactory task performance, due to binary spike feature maps of the SNN will result in limited expressiveness and large accuracy degradation compared with full-precision feature maps of the ANN",
>
> **A4**: Thanks for the reminder. We will further recheck and polish our paper in the final version.

---

> > ### Comment · Reviewer_bdtv · 2024-08-11
> > **response**
> >
> > Thank you for your feedback.
> > The authors have worked to improve the manuscript's readability.
> > I am inclined to increase my score to 7.

---

> > > ### Author Response · Authors · 2024-08-12
> > > **Thanks**
> > >
> > > Thanks very much for your reply and recognition. We are happy to see that your concerns have been addressed.

---

### Decision · Program_Chairs · 2024-09-25

**Decision:**

Accept (poster)

**Comment:**

This paper has received positive reviews. All reviewers agree that the paper has novelty, clarity and boosts the effectiveness of training for SNNs.